# *Ldlr*-Deficient Mice with an Atherosclerosis-Resistant Background Develop Severe Hyperglycemia and Type 2 Diabetes on a Western-Type Diet

**DOI:** 10.3390/biomedicines10061429

**Published:** 2022-06-16

**Authors:** Weibin Shi, Jing Li, Kelly Bao, Mei-Hua Chen, Zhenqi Liu

**Affiliations:** 1Department of Radiology and Medical Imaging, University of Virginia, Charlottesville, VA 22908, USA; jjeileen2006@hotmail.com (J.L.); kqb2j@virginia.edu (K.B.); mc2xc@hscmail.mcc.virginia.edu (M.-H.C.); 2Department of Medicine, University of Virginia, Charlottesville, VA 22908, USA; zl3e@hscmail.mcc.virginia.edu

**Keywords:** low-density lipoprotein receptor, apolipoprotein E, hyperglycemia, type 2 diabetes, Western diet

## Abstract

*Apoe*^-/-^ and *Ldlr*^-/-^ mice are two animal models extensively used for atherosclerosis research. We previously reported that *Apoe*^-/-^ mice on certain genetic backgrounds, including C3H/HeJ (C3H), develop type 2 diabetes when fed a Western diet. We sought to characterize diabetes-related traits in C3H-*Ldlr*^-/-^ mice through comparing with C3H-*Apoe*^-/-^ mice. On a chow diet, *Ldlr*^-/-^ mice had lower plasma total and non-HDL cholesterol levels but higher HDL levels than *Apoe*^-/-^ mice. Fasting plasma glucose was much lower in *Ldlr*^-/-^ than *Apoe*^-/-^ mice (male: 122.5 ± 5.9 vs. 229.4 ± 17.5 mg/dL; female: 144.1 ± 12.4 vs. 232.7 ± 6.4 mg/dL). When fed a Western diet, *Ldlr*^-/-^ and *Apoe*^-/-^ mice developed severe hypercholesterolemia and also hyperglycemia with fasting plasma glucose levels exceeding 250 mg/dL. Both knockouts had similar non-HDL cholesterol and triglyceride levels, and their fasting glucose levels were also similar. Male *Ldlr*^-/-^ mice exhibited greater glucose tolerance and insulin sensitivity compared to their *Apoe*^-/-^ counterpart. Female mice showed similar glucose tolerance and insulin sensitivity though *Ldlr*^-/-^ mice had higher non-fasting glucose levels. Male *Ldlr*^-/-^ and *Apoe*^-/-^ mice developed moderate obesity on the Western diet, but female mice did not. These results indicate that the Western diet and ensuing hyperlipidemia lead to the development of type 2 diabetes, irrespective of underlying genetic causes.

## 1. Introduction

Diabetes is a chronic health condition featured by fasting hyperglycemia (elevations in blood sugar). Type 2 diabetes (T2D) is the major form of diabetes accounting for 90–95% of all diabetes and constitutes one of the major public health problems in the U.S. and globally. Thirty-five million (13.5%) Americans aged 18 years or older are believed to have T2D [1]. Globally, 463 million adults are estimated to be living with T2D [2]. Additionally, 92 million American adults (37.6%) have prediabetes [1], a condition that carries the risk of contracting T2D. Diabetic patients experience increased rates of macrovascular complications, including more than twice the rates of coronary artery disease and stroke [3,4], and microvascular retinopathy and chronic kidney disease [5,6].

Dyslipidemia, which comprises elevated triglyceride and LDL cholesterol levels and reduced HDL cholesterol levels, often occurs together with hyperglycemia as integral components of T2D and the metabolic syndrome, with the latter also including abdominal obesity and hypertension. Accumulating evidence suggests that dyslipidemia may play a causal role in the pathogenesis of T2D. Prospective studies show that individuals with high triglyceride and LDL cholesterol levels have an increased risk of developing T2D [7,8,9,10]. Strong correlations between plasma lipid and glucose levels have been observed in segregating F2 populations derived from *Apoe^-/-^* mouse strains [11,12,13]. The strong evidence supporting a causal role for dyslipidemia in diabetes comes from genetic studies of rare mutations in *ABCA1* [14], LIPE [15], *LPL* [16], and *LRP6* [17], showing that these lipid genes are also linked to hyperglycemia or diabetes. However, contradictory findings concerning the relationship between dyslipidemia and diabetes are also noted: Hyperlipidemic patients receiving lipid-lowering therapy with PCSK9 inhibitory antibodies or statins show an increased incidence of new-onset diabetes [18,19,20]. Patients with heterozygous familial hypercholesterolemia caused by mutations in *LDLR*, *APOB*, *APOE,* and *PCSK9* are less vulnerable to diabetes [21,22].

*Apoe*^-/-^ and *Ldlr*^-/-^ mice are two widely used rodent models of dyslipidemia and atherosclerosis. On a chow diet, *Apoe*^-/-^ mice have total plasma cholesterol levels of 300 to 500 mg/dL mainly from elevations of VLDL and chylomicron remnants, and *Ldlr*^-/-^ mice have cholesterol levels of 200 to 300 mg/mL due to an accumulation of LDL [23,24,25,26]. When fed a Western diet, both *Apoe*^-/-^ and *Ldlr*^-/-^ mice develop severe hypercholesterolemia, with total cholesterol levels of ~1000 mg/dL [23,25,26]. We found that *Apoe*^-/-^ mice on certain genetic backgrounds such as C57BL/6 (B6), C3H/HeJ (C3H), and SWR/J develop significant hyperglycemia and T2D with fasting plasma glucose exceeding 250 mg/dL when fed a Western-type diet [27,28]. Thus, we reasoned that *Ldlr*^-/-^ mice would develop hyperglycemia and diabetes on a Western diet due to the ensuing severe dyslipidemia. To test this hypothesis, we characterized diabetes-related phenotypes in *Ldlr*^-/-^ mice by comparing with *Apoe*^-/-^ mice.

## 2. Materials and Methods

### 2.1. Mice

C3H-*Apoe*^-/-^ mice and C3H-*Ldlr*^-/-^ mice at N10 or more backcrossed generations were generated in our laboratory using the classical congenic breeding strategy [28]. Mice of both sexes were weaned at 3 weeks of age onto a chow diet containing 19% protein, 5% fat, 5% crude fiber, and 58% of the calories from carbohydrates (Teklad LM-485, Envigo, Indianapolis, IN, USA). At 6 weeks of age, one group was switched onto a Western diet containing 21% fat, 34.1% sucrose, 0.15% cholesterol, and 19.5% casein by weight (TD 88137, Envigo) and maintained on the diet for 12 weeks. The other group remained on a rodent chow diet throughout the entire experimental period. The animals were housed in a pathogen-free facility with a 12 h light–12 h dark cycle, consistent temperature (22 °C), and 50% humidity. Animal care and experimentation were carried out according to the current National Institutes of Health guidelines and approved by the University of Virginia Animal Care and Use Committee (Protocol # 3109). The reporting in this study follows the recommendations in the ARRIVE guidelines.

### 2.2. Measurements of Plasma Lipids, Glucose and Insulin

Mice were bled twice: once before and once at the end of the Western diet feeding. The animals were fasted overnight before retro-orbital blood was collected under isoflurane anesthesia. Plasma levels of total cholesterol, HDL cholesterol, and triglyceride were measured using cholesterol reagent from Stanbio Laboratory, HDL precipitating reagent from FUJIFILM Wako Diagnostics, and triglyceride reagent from Thermo DMA (Louisville, CO, USA) as reported in [11]. Plasma was diluted 3x with H_2_O for measurement of total cholesterol concentrations in mice fed the Western diet. Plasma glucose concentrations were determined using a Sigma assay kit (Cat. # GAHK20) as reported in [11]. Plasma was diluted 3x for mice fed the Western diet but not diluted for chow-fed mice. Plasma insulin concentrations were measured with an ultra-sensitive ELISA kit from Crystal Chem INC (Elk Grove Village, IL; Cat. # 90080) according to the manufacturer’s instructions.

### 2.3. Glucose Tolerance Test (GTT) and Insulin Tolerance Test (ITT)

GTT and ITT were performed as we described [27,28]. Briefly, overnight-fasted mice were injected intraperitoneally (IP) with 1 mg of glucose in 0.9% saline per gram of body weight in a volume of <0.3 mL. Blood glucose levels were measured with an UltraTouch glucometer using blood taken from cut tail tip at 0, 10, 20, 30, 60, 90, and 120 min after injection. ITT was performed under a non-fasting condition by IP injection with 0.75 U insulin in 0.9% saline per kg of body weight. Blood glucose was measured as above at 0, 15, 30, 45, and 60 min after insulin injection.

### 2.4. Statistical Analysis

Values were expressed as means ± SE, with *n* indicating the number of animals. The distributions of trait values were assessed for normality by examining skewness, kurtosis, and Q–Q plot as reported in [29]. Student’s t test and ANOVA (Analysis of Variance) were used for determining statistical significance between two or more groups. When the *p*-value for ANOVA was statistically significant, Dunnett’s correction was applied. Differences were considered statistically significant at *p* < 0.05.

## 3. Results

### 3.1. Fasting Plasma Lipid Levels

Both C3H-*Apoe*^-/-^ and C3H-*Ldlr*^-/-^ mice developed spontaneous hypercholesterolemia on a chow diet, and it was more severe in the former (Figure 1). Total cholesterol levels of C3H-*Ldlr*^-/-^ mice were significantly lower than those of C3H-*Apoe*^-/-^ mice (male: 285.8 ± 10.1 vs. 344.1 ± 24.6 mg/dL; female: 302.4 ± 14.6 vs. 419.2 ± 11.8 mg/dL; *p* ≤ 0.013; *n* = 11 to 29). On the Western diet, both genotypes developed severe hypercholesterolemia. The total cholesterol level was significantly lower in male C3H-*Ldlr*^-/-^ mice than in the C3H-*Apoe*^-/-^ counterparts (719.1 ± 46.2 vs. 917.0 ± 43.1 mg/dL; *p* = 0.035; *n* = 4 to 12). Female C3H-*Ldlr*^-/-^ mice did not differ significantly in total cholesterol level from the C3H-*Apoe*^-/-^ counterparts (904.2 ± 41.3 vs. 1075.3 ±106.5 mg/dL; *p* = 0.09; *n* = 6 to 11).

Male C3H-*Ldlr*^-/-^ mice had significantly higher HDL cholesterol levels than the C3H-*Apoe*^-/-^ counterparts on both chow (99.1 ± 5.6 vs. 57.0 ± 2.1 mg/dL; *p* = 0.0023; *n* = 11 to 29) and Western diets (94.7 ± 5.4 vs. 50.1 ± 4.3 mg/dL; *p* = 0.00057; *n* = 4 to 12) (Figure 2). Though statistically insignificant, female C3H-*Ldlr*^-/-^ mice also had higher HDL cholesterol levels than female C3H-*Apoe*^-/-^ mice on either chow (76.9 ± 2.1 vs. 53.0 ± 7.1 mg/dL; *p* = 0.058; *n* = 11 to 29) or the Western diet (44.5 ± 6.6 vs. 31.6 ± 2.1 mg/dL; *p* = 0.10; *n* = 6 to 11).

Non-HDL cholesterol levels of *Ldlr*^-/-^ mice were significantly lower than those of the *Apoe*^-/-^ counterparts on the chow diet for both males (119.2 ± 8.7 vs. 420.1 ± 50.1 mg/dL; *p* = 0.0096; *n* = 4 to 18) and females (102.5 ± 8.7 vs. 303.6 ± 24.4 mg/dL; *p* = 0.0002; *n* = 4–9) (Figure 2A,B). On the Western diet, non-HDL levels were dramatically elevated in *Ldlr*^-/-^ and *Apoe*^-/-^ mice of both sexes, exceeding 800 mg/dL (Figure 2C,D). No significant difference was found between the two knockouts (*p* > 0.2).

C3H-*Ldlr*^-/-^ and C3H-*Apoe*^-/-^ mice had similar plasma triglyceride levels except for males on the chow diet, with the former being significantly lower (100.1 ± 4.2 vs. 150.4 ± 15.3 mg/dL; *p* = 0.00018; *n* = 4 to 29).

### 3.2. Fasting Plasma Glucose Levels

On the chow diet, both male and female C3H-*Ldlr*^-/-^ mice had 40~50% lower fasting glucose levels than the C3H-*Apoe*^-/-^ counterparts (male: 122.5 ± 5.9 vs. 229.4 ± 17.5 mg/dL; *p* = 2.0 × 10^−8^; female: 144.1 ± 12.4 vs. 232.7 ± 6.4 mg/dL; *p* = 1.6 × 10^−6^; *n* = 11 to 29) (Figure 3). On the Western diet, both C3H-*Ldlr*^-/-^ and C3H-*Apoe*^-/-^ mice developed significant hyperglycemia, with fasting glucose levels exceeding 250 mg/dL. Male C3H-*Ldlr*^-/-^ mice had a lower plasma glucose level (304.7 ± 21.8 vs. 398.1 ± 46.4 mg/dL; *p* = 0.061; *n* = 4 to 13), while female C3H-*Ldlr*^-/-^ mice had a higher plasma glucose level when compared to the C3H-*Apoe*^-/-^ counterparts (381.7 ± 25.4 vs. 297.4 ± 31.1 mg/dL; *p* = 0.06; *n* = 4 to 7). Compared to the chow diet, the Western diet significantly elevated plasma glucose levels of C3H-*Ldlr*^-/-^ and C3H-*Apoe*^-/-^ mice (*p* ≤ 0.0055).

### 3.3. Glucose Tolerance Test (GTT) and Insulin Tolerance Test (ITT)

Intraperitoneal GTT and ITT were performed on both male and female C3H-*Ldlr*^-/-^ and C3H-*Apoe*^-/-^ mice fed the Western diet. In response to injected glucose, blood glucose levels rose quickly to the peak at the 20th min for male C3H-*Ldlr*^-/-^ mice and the 30th min for male *Apoe*^-/-^ mice (Figure 4). Then, glucose levels fell gradually and returned to pre-injection levels by the 120th min. Compared to C3H-*Apoe*^-/-^ counterparts, male C3H-*Ldlr*^-/-^ mice displayed significant glucose tolerance (*p* < 0.001; *n* = 4 to 17), having lower glucose levels at the 10th, 20th, and 30th min. For female mice, blood glucose levels quickly reached the peak at the 10th min and then fell gradually. There was no significant difference in glucose tolerance between female C3H-*Ldlr*^-/-^ and C3H-*Apoe*^-/-^ mice (*p* = 0.25; *n* = 4 to 17).

In response to injected insulin, male C3H-*Ldlr*^-/-^ mice showed a gradual fall in blood glucose levels till the 30th min, which remained low till the 45th min and recovered at the 60th min (Figure 5A). In contrast, male C3H-*Apoe*^-/-^ mice showed an increase in blood glucose levels at the 15th min though glucose levels were comparable to the levels of C3H-*Ldlr*^-/-^ mice from the 30th to 60th min (16% reduction from the basal) (Figure 5C). Female C3H-*Ldlr*^-/-^ mice showed a more obvious fall in blood glucose levels than female C3H-*Apoe*^-/-^ mice (47% vs. 18% decrease from the basal level) (Figure 5B,D). In addition, the basal non-fasting blood glucose level (at 0 min) was significantly higher in female C3H-*Ldlr*^-/-^ mice than in the C3H-*Apoe*^-/-^ counterparts (165.4 ± 13.7 vs. 118.5 ± 5.4 mg/dL; *p* = 0.006; *n* = 4 to 20).

### 3.4. Plasma insulin Concentration

Fasting plasma insulin levels were measured for female C3H-*Ldlr*^-/-^ and C3H-*Apoe*^-/-^ mice fed the Western diet. C3H-*Ldlr*^-/-^ mice had higher insulin levels than the C3H-*Apoe*^-/-^ mice (0.622 ± 0.121 vs. 0.487 ± 0.074 ng/mL), although the difference was not statistically significant (*p* = 0.33; *n* = 7 to 13) (Figure 6).

### 3.5. Body Weight

At 3 months of age on the chow diet, either male or female C3H-*Ldlr*^-/-^ mice had a body weight similar to that of the C3H-*Apoe*^-/-^ mice (male: 23.1 ± 0.9 vs. 23.2 ± 0.7 g; female: 17.2 ± 0.3 vs. 18.1 ± 0.5 g) (Figure 7). After being fed the Western diet for 12 weeks, C3H-*Ldlr*^-/-^ and C3H-*Apoe*^-/-^ mice also had a similar body weight (male: 33.2 ± 0.8 vs. 35.0 ± 0.8 g; female: 23.1 ± 1.2 vs. 23.8 ± 0.8 g). Males had significantly heavier body weight than females for both the C3H-*Ldlr*^-/-^ and C3H-*Apoe*^-/-^ mice fed either chow or a Western diet (*p* < 0.05; *n* = 4 to 26).

## 4. Discussion

In this study, we characterized type 2 diabetes-associated phenotypes in C3H-*Ldlr*^-/-^ mice through a comparison with C3H-*Apoe*^-/-^ mice, which develop type 2 diabetes on a Western diet [28]. Our results here indicate that Western diet consumption and ensuing hyperlipidemia lead to the development of type 2 diabetes in hyperlipidemic mice, irrespective of underlying genetic causes. On a chow diet, plasma total and non-HDL cholesterol levels were significantly lower in C3H-*Ldlr*^-/-^ mice than in C3H-*Apoe*^-/-^ mice and so were the fasting glucose levels. When fed a Western diet, C3H-*Ldlr*^-/-^ mice developed just as severe hyperlipidemia as C3H-*Apoe*^-/-^ mice, and the magnitude of hyperglycemia was similar. Male C3H-*Ldlr*^-/-^ mice were more tolerant to glucose loading and more sensitive to insulin than male C3H-*Apoe*^-/-^ mice, while female C3H-*Ldlr*^-/-^ mice were just as tolerant to glucose and sensitive to insulin as female C3H-*Apoe*^-/-^ mice. Moreover, male but not female C3H-*Ldlr*^-/-^ and C3H-*Apoe*^-/-^ mice developed moderate obesity on the Western diet.

A major finding of this study is that C3H-*Ldlr*^-/-^ mice developed type 2 diabetes with fasting plasma glucose levels ranging from 300 to 400 mg/dL after 12 weeks of a Western diet. Diabetes is defined by fasting hyperglycemia. In humans, fasting blood glucose levels of ≥ 126 mg/dL are considered diabetic. For mice, plasma instead of whole blood is often used for glucose measurements. Fasting plasma glucose exceeding 250 mg/dL is considered diabetic for mice [30]. Thus, both male and female C3H-*Ldlr*^-/-^ mice developed diabetes on the Western diet. Compared to the C3H-*Apoe*^-/-^ males, the C3H-*Ldlr*^-/-^ males had lower total and non-HDL cholesterol and higher HDL cholesterol levels, and their plasma glucose levels were also lower. Similarly, C3H-*Ldlr*^-/-^ and C3H-*Apoe*^-/-^ females had comparable total, non-HDL, and HDL cholesterol levels, and their fasting glucose levels were comparable. These results hint at the significance of lipid profiles in determining blood glucose homeostasis.

Schreyer et al. [31] observed a time-dependent increase in fasting glucose levels of B6-*Ldlr*^-/-^ mice but not in B6-*Apoe*^-/-^ mice within 16 weeks on a diabetogenic diet containing 35.5% fat, 36.6% carbohydrates, and no cholesterol, although the increase did not meet the threshold of 250 mg/dL for the diagnosis of diabetes. Phillips et al. [32] found that only the Western diet but not a diabetogenic diet containing 13% fat and 67% carbohydrates raised blood glucose level of B6-*Apoe*^-/-^ mice.

C3H-*Ldlr*^-/-^ mice developed milder hypercholesterolemia on the chow diet than C3H-*Apoe*^-/-^ mice, having lower total and non-HDL cholesterol levels but higher HDL cholesterol levels. These results are in line with what has been found with the two knockouts on the C57BL/6 background [33]. Lipid profiles vary by strains [28], but the distinctions between *Ldlr*^-/-^ and *Apoe*^-/-^ mice in total and HDL and non-HDL cholesterol levels remain on the C3H background. What is more distinct are the fasting glucose levels, which were much lower in the C3H-*Ldlr*^-/-^ mice than in the C3H-*Apoe*^-/-^ mice of both sexes. Differential fasting glucose levels of the two knockouts were in parallel with their differing lipid profiles, with lower non-HDL and higher HDL cholesterol levels being linked with lower fasting glucose levels. Consistently, in multiple F2 cohorts derived from *Apoe*^-/-^ mouse strains, plasma glucose levels are positively correlated with non-HDL cholesterol levels [11,12,13] and inversely correlated with HDL levels [13].

Although both male and female C3H-*Ldlr*^-/-^ and C3H-*Apoe*^-/-^ mice developed significant hyperglycemia on the Western diet, only C3H-*Apoe*^-/-^ males exhibited noticeable glucose intolerance. Indeed, male C3H-*Apoe*^-/-^ mice showed an increase in blood glucose concentration after intraperitoneal glucose injection. As there was little fall in blood glucose concentration after insulin injection, these mice developed strong insulin resistance on the Western diet. Thus, insulin resistance should be responsible in part for the glucose intolerance and hyperglycemia observed in male *Apoe*^-/-^ mice. Obesity and hyperlipidemia were two obvious factors contributing to the development of insulin resistance. Both C3H-*Ldlr*^-/-^ and C3H-*Apoe*^-/-^ males fed the Western diet developed moderate obesity as evidenced by substantial increases in body weight when compared to age-matched chow-fed mice (33.2 ± 0.8 vs. 24.1 ± 1.0 g for C3H-*Ldlr*^-/-^ mice, 35.0 ± 0.8 vs. 26.2 ± 1.2 g for C3H-*Apoe*^-/-^ mice). However, the finding that the two knockouts had similar body weight but male *Ldlr*^-/-^ mice did not develop obvious insulin resistance suggests that other factors than obesity also contributed to glucose intolerance and insulin resistance of male C3H-*Apoe*^-/-^ mice. The consumption of a high-fat diet leads to hyperlipidemia, which is a major driver of oxidative stress and systemic chronic inflammation [34,35]. Besides its role in lipid metabolism, ApoE has antioxidant effects and suppresses inflammation in the body [36]. *Apoe*^-/-^ mice show enhanced responses to oxidative stress and inflammatory stimuli compared to *Ldlr*^-/-^ mice [37,38]. The present finding that female C3H-*Apoe*^-/-^ mice also exhibited an increased insulin resistance relative to the *Ldlr*^-/-^ counterparts supports the speculation on enhanced oxidative stress and inflammatory responses in *Apoe*^-/-^ mice.

Unlike their male counterparts and most other mouse models of type 2 diabetes that develop significant obesity [39], female C3H-*Ldlr*^-/-^ and C3H-*Apoe*^-/-^ mice had no overweight or obesity. Despite the absence of obesity, female C3H-*Ldlr*^-/-^ and C3H-*Apoe*^-/-^ mice developed just as severe hyperglycemia as their male counterparts. In humans, although most type 2 diabetic patients are obese or overweight, a fraction of patients have normal body weight [40]. On the Western diet, the two female knockouts had similar lipid profiles, including total, HDL, non-HDL cholesterol, and triglyceride. Interestingly, they also had comparable fasting plasma glucose and insulin levels. This coincidence supports the significance of lipid profiles in the development of type 2 diabetes. There are quantitative but not qualitative differences in plasma insulin levels between male and female hyperlipidemic mice fed a Western diet [41]. For more than 100 inbred strains examined, male mice have higher plasma insulin levels than female mice. Previous studies showed that a high-fat diet had little or no influence on plasma glucose levels of wild-type B6 mice [42,43], probably due to the moderateness in the increase in plasma lipid levels. This speculation appears applicable to wild-type C3H mice in that they only developed modest hyperlipidemia (plasma cholesterol level: 181 ± 14 mg/dL) and showed a modest increase in fasting glucose levels on the Western diet (148 ± 14 vs. 106 ± 11 mg/dL) (Appendix A).

In humans, *APOE* and *LDLR* polymorphisms have been associated with obesity [44]. We previously observed a deficit in growth and weight gain during postnatal development of *Apoe*^-/-^ mice as compared to wild-type mice [45]. Male *Apoe*^-/-^ mice showed resistance to Western diet-induced obesity [46]. *Ldlr*^-/-^ mice also showed a decreased susceptibility to Western diet-induced obesity due to an increased thermogenesis [47]. Here we also observed that male *Ldlr*^-/-^ mice had a smaller body weight in comparison to male *Apoe*^-/-^ mice on the Western diet.

## 5. Conclusions

*Ldlr* deficiency results in increased levels of ApoB-100 containing LDL due to both increased hepatic lipoprotein production and impaired clearance [48,49], while ApoE deficiency leads to the accumulation of ApoB48 containing chylomicron or VLDL remnants due to a delayed clearance into hepatocytes by the LDL receptor, LDL receptor-related protein 1, and cell surface heparan sulphate proteoglycans [50]. Despite different lipoprotein profiles, *Ldlr*^-/-^ mice and *Apoe*^-/-^ mice developed diet-induced type 2 diabetes that is independent of obesity. This finding provides direct evidence for the significance of a Western-type diet and its induced hyperlipidemia in the etiology of type 2 diabetes. The current observations are not specific to C3H mice because *Ldlr*^-/-^ mice with the B6;129 genetic background also developed hyperglycemia on the Western diet, which was alleviated after mice were switched to a low-fat diet [51]. We previously observed that the Western diet-induced increases in plasma glucose and triglyceride levels of *Apoe*^-/-^ mice are time-dependent within the 12-week feeding period [52,53], while the current study only examined one time point on the Western diet. Another limitation is that wild-type C3H mice were not included in this study. In addition, sample sizes in some experiments were small despite the fact that the two knockouts were inbred and tended to have small intragroup variation in the phenotypes tested. Nevertheless, there is no doubt that the two mouse models are useful for investigating type 2 diabetes that is related or unrelated to obesity.

## Figures and Tables

**Figure 1 biomedicines-10-01429-f001:**
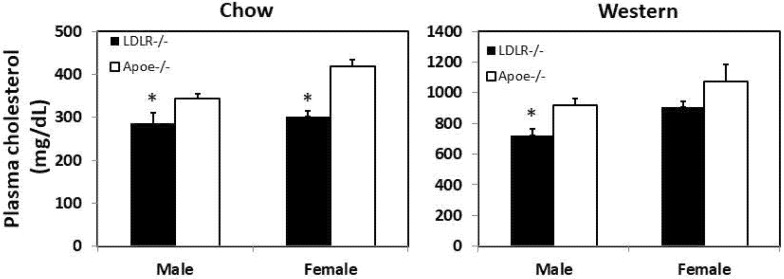
Fasting plasma levels of total cholesterol in male and female C3H-*Apoe*^-/-^ and C3H-*Ldlr*^-/-^ mice when fed a chow (left panel) or Western diet (right panel). Results are means ± SE of 4 to 29 mice per group. * *p* < 0.05 vs. *Apoe*^-/-^ mice.

**Figure 2 biomedicines-10-01429-f002:**
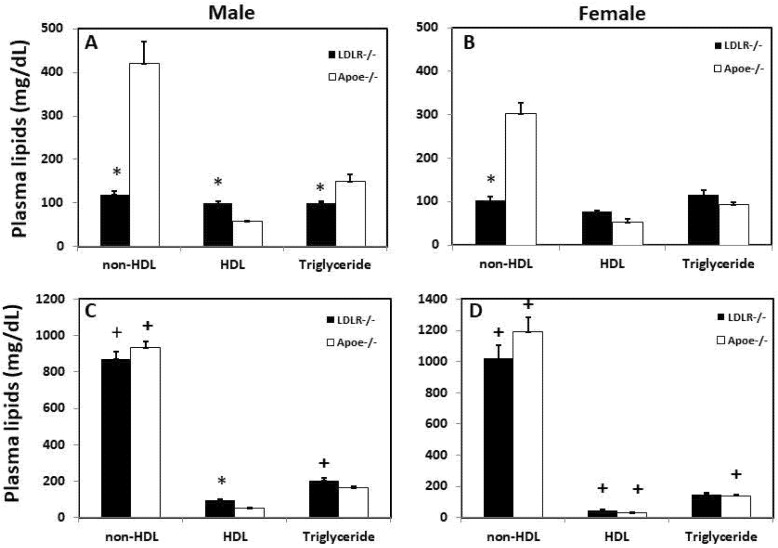
Fasting plasma levels of non-HDL, HDL cholesterol, and triglyceride in male and female C3H-*Apoe*^-/-^ and C3H-*Ldlr*^-/-^ mice fed a chow (**A**,**B**) or Western diet (**C**,**D**). Results are means ± SE of 4 to 29 mice per group. * *p* < 0.05 vs. *Apoe*^-/-^ mice; + *p* < 0.05 vs. chow diet.

**Figure 3 biomedicines-10-01429-f003:**
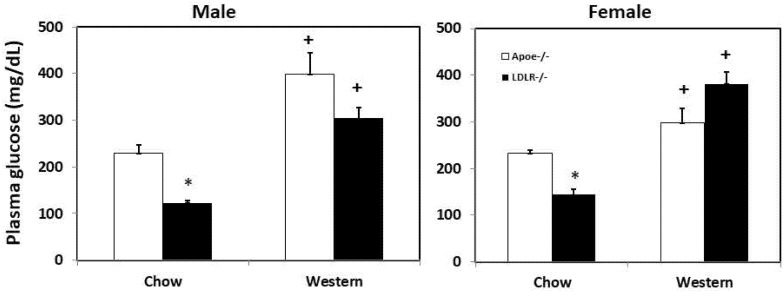
Fasting plasma glucose levels of male and female C3H-*Apoe*^-/-^ and C3H-*Ldlr*^-/-^ mice fed a chow or Western diet. Results are means ± SE of 4 to 29 mice per group. * *p* < 0.05 vs. *Apoe*^-/-^ mice and ^+^ *p* < 0.05 vs. chow diet.

**Figure 4 biomedicines-10-01429-f004:**
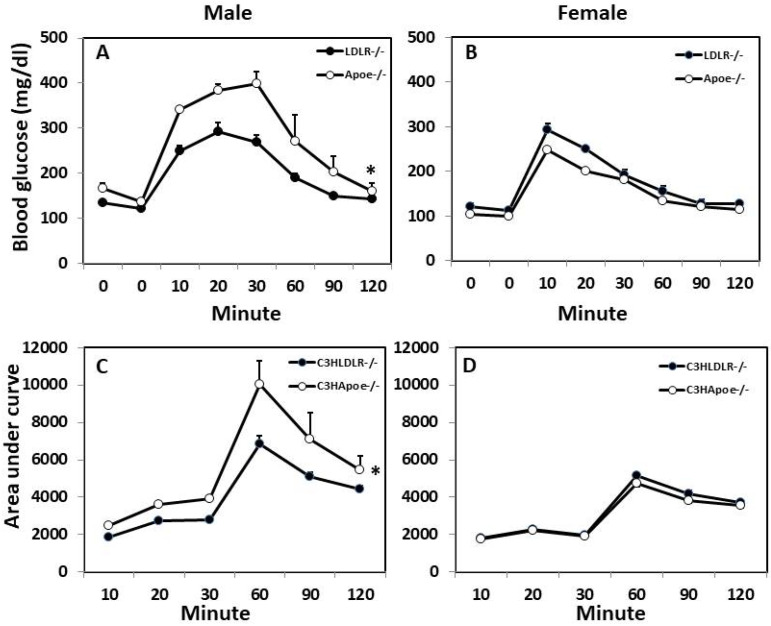
Glucose tolerance test (GTT) (**A**,**B**) and calculated area under the curve (**C**,**D**) for male and female C3H-*Apoe*^-/-^ and C3H-*Ldlr*^-/-^ mice fed a Western diet. Overnight-fasted mice were intraperitoneally injected with 1 g glucose per kg body weight. Blood glucose concentrations were determined with a glucometer using blood taken from cut tail tips at the indicated time points. Values are means ± SE of 4 to 17 mice per group. * *p* < 0.05 vs. *Apoe*^-/-^ mice.

**Figure 5 biomedicines-10-01429-f005:**
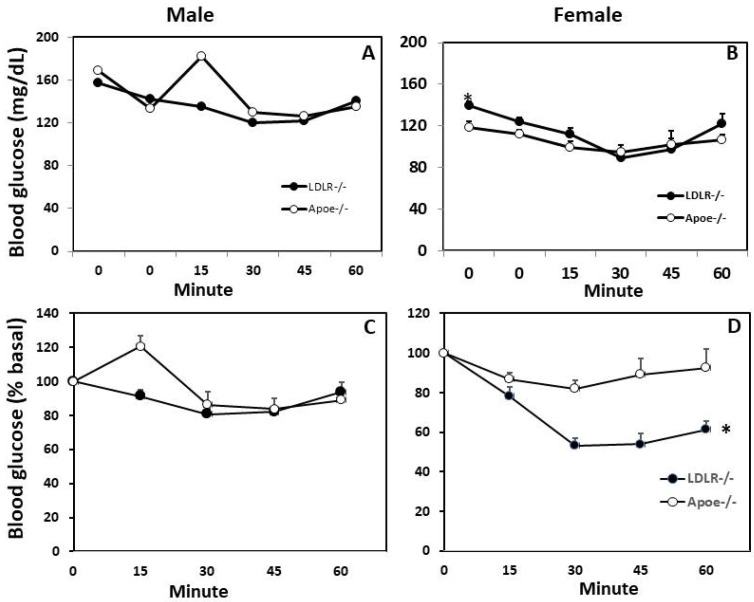
Insulin tolerance test (ITT) for male and female C3H-*Apoe*^-/-^ and C3H-*Ldlr*^-/-^ mice fed a Western diet. Non-fasted mice were intraperitoneally injected with 0.75 U/kg of insulin. Blood glucose concentrations were determined with a glucometer using blood taken from cut tail tips at the indicated time points (**A**,**B**). Values are means ± SE of 4 to 20 mice per group. * *p* < 0.05 vs. *Apoe*^-/-^ mice. (**C**,**D**), Glucose concentrations were expressed as % of the basal.

**Figure 6 biomedicines-10-01429-f006:**
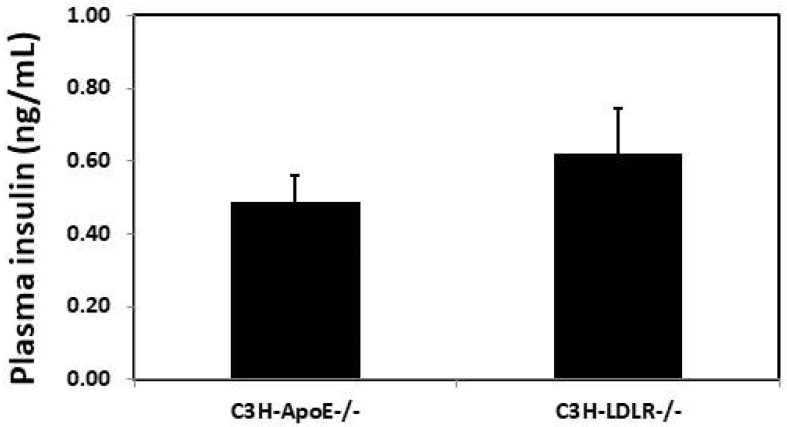
Plasma insulin levels of female C3H-*Apoe*^-/-^ and C3H-*Ldlr*^-/-^ mice fed a Western diet. Blood samples were collected after mice were fasted overnight. Values are means ± SE of 7 or 13 mice per group.

**Figure 7 biomedicines-10-01429-f007:**
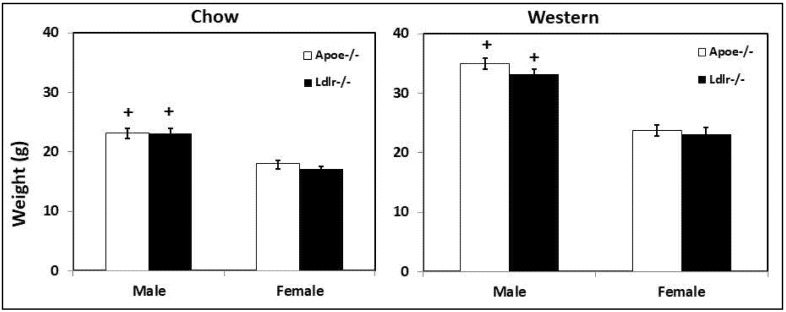
Body weight (g) of male and female C3H-*Apoe*^-/-^ and C3H-*Ldlr*^-/-^ mice fed a chow or a Western diet. Chow-fed mice were weighed at 3 months of age, and Western diet-fed mice were weighed after being euthanized. Results are means ± SE of 4 to 26 mice. ^+^ *p* < 0.05 vs. female mice.

## Data Availability

All data reported in the article are included in the Appendix A and also accessible via 10.6084/m9.figshare.20032229.

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
