# Peer review of "Ldlr*-Deficient Mice with an Atherosclerosis-Resistant Background Develop Severe Hyperglycemia and Type 2 Diabetes on a Western-Type Diet"

_biomedicines, 2022, doi:10.3390/biomedicines10061429_

Round 1
Reviewer 1 Report
An interesting topic. The study concluded that both male and female C3H-Ldlr -/- mice developed diabetes on the Western diet.
General comments:
In the abstract is concluded that: These results indicate that Western diet and ensuing hyperlipidaemia lead to development of type 2 diabetes, irrespective of its underlying causes. The title of this manuscript is diffused in the abstract text; it should be emphasized.
The reviewer suggests authors to revise the manuscript based on some of the suggestions below:
Page 5.: Materials and methods – Mice: weight of the animals at the experimental starting, the composition of animal groups (males, females), how they were housed, ambient conditions ….the animal experiment design should be explained.
The manuscript is not yet satisfactory for publishing.
Standard chow diet – composition
Western diet – fat %?
Page 6.: 1mg of glucose /g BW – dissolved in NaCl? How many ml were injected?
Page 7.: figure 1: Results are means ± SE of 4 to 29 mice per group. Meaning of 4 to 29 mice /group
Page 10.: figure 4 .: Values are means ± SE of 3 to 17 mice per group. Meaning of 4 to 29 mice /group. See fig 5, fig 7
Page 11.: figure 6.: 7 or 13 mice per group.
Author Response
Comment: In the abstract is concluded that: These results indicate that Western diet and ensuing hyperlipidaemia lead to development of type 2 diabetes, irrespective of its underlying causes. The title of this manuscript is diffused in the abstract text; it should be emphasized.
Response: Revised. A word “genetic” is added to narrow the scope of conclusions.
Comment: Page 5.: Materials and methods – Mice: weight of the animals at the experimental starting, the composition of animal groups (males, females), how they were housed, ambient conditions ….the animal experiment design should be explained.
Response: Amended. Mice were only weighed at the end of Western diet feeding.
Comment: Standard chow diet – composition
Response: Amended.
Comment: Western diet – fat %?
Response: 21% fat by weight.
Comment: Page 6.: 1mg of glucose /g BW – dissolved in NaCl? How many ml were injected?
Response: Clarified. Glucose was in 0.9% saline. The amount injected varies depending on body weight. On average, 0.17 ml was used for female mice and 0.27 ml for male mice.
Comment: Page 7.: figure 1: Results are means ± SE of 4 to 29 mice per group. Meaning of 4 to 29 mice /group.
Response: Yes.
Comment: Page 10.: figure 4 .: Values are means ± SE of 3 to 17 mice per group. Meaning of 4 to 29 mice /group. See fig 5, fig 7
Response: Values are means ± SE of 4 to 17 mice per group. We tried to do ITT or GTT by including both C3H-Apoe-/- and C3H-LDLR-/- mice, but it was hard to generate sufficient mice from a small breeding colony.
Comment: Page 11.: figure 6.: 7 or 13 mice per group.
Response: yes, the number of mice in each group varies from experiment to experiment.
Reviewer 2 Report
This work presents the differences in metabolic alterations in Ldlr-/ and Apoe-/ mice under normal and Western diet. Finally, the authors conclude that the Western diet produces moderate obesity in males and type 2 diabetes in all animals, independently of their underlying causes.
The paper presents results based on two diets and two animal models from the same mouse background. The work shows important differences in metabolic alterations between the two mouse models and by sex. The work focuses on the consequences of the Western diet on the development of diabetes. The main problem of the work is the absence of control (wild type) mice. This control may show whether the lipid alterations present in the mouse models may be relevant in the development of diabetes as both knockout mice have increased cholesterol levels and lower HDL levels. Western diet is known to generate diabetes in mice but it might be interesting to know if dyslipidemia can modify this.
Other comments:
- How do the authors define type 2 diabetes in mice?
- Have the authors analyzed glucose levels over the months on the diet? When did the glucose alterations start?
- How many mice were in each group?
Author Response
Comment: The paper presents results based on two diets and two animal models from the same mouse background. The work shows important differences in metabolic alterations between the two mouse models and by sex. The work focuses on the consequences of the Western diet on the development of diabetes. The main problem of the work is the absence of control (wild type) mice. This control may show whether the lipid alterations present in the mouse models may be relevant in the development of diabetes as both knockout mice have increased cholesterol levels and lower HDL levels. Western diet is known to generate diabetes in mice but it might be interesting to know if dyslipidemia can modify this.
Response: Wild-type mice were not included as they only show modest rise in blood glucose levels and do not develop significant hyperglycemia (Scientific Reports 2019; 9: 19556. J Diabetes 2015;7:74. Parks et al., 2015, Cell Metabolism 21, 334). Our observation of 4 female C3H/HeJ mice also show that these mice do not develop significant hyperplasia after 12 weeks on a Western diet (plasma levels of 148 ± 14 mg/dL) and nor overweight (21 ± 0.6 g), while female C3H-Apoe-/- mice had plasma glucose levels of 312 ± 31 mg/dL and body weight (21 ± 0.2 g) (data provided under “C3H-wild type” in Supplemental materials). We have also addressed this concern in Discussion.
Comment:- How do the authors define type 2 diabetes in mice?
Response: Diabetes is defined by fasting hyperglycemia. In humans, fasting blood glucose levels of ≥ 126 mg/dL are considered diabetic. For mice, plasma instead of whole blood is often used for glucose measurements. Fasting plasma glucose exceeding 250 mg/dL is considered diabetic for mice [31].
Comment: - Have the authors analyzed glucose levels over the months on the diet? When did the glucose alterations start?
Response: We previously reported time-dependent rises in plasma glucose levels of B6-Apoe-/- and C3H-Apoe-/- mice within 12 weeks of Western diet (Physiol Genomics 2012; 44: 345. BMC Endocr Disord 2015;15:13). Plasma triglyceride levels also show a time-dependent rise within 12 weeks of Western diet (BMC Endocr Disord 2015;15:13).
Comment: - How many mice were in each group?
Response: There were 10 to 30 mice per group, but some assays were not run for all mice due to small volumes of blood that could be obtained from mice.
Reviewer 3 Report
The authors provide new information on the ldlr-/- and apoe-/- models and how the two compare.
What is missing from the paper is any wildtype control mice as a reference. Especially when the authors claim that ldlr-/- and apoe-/- mice do not develop overweight when fed a western diet, yet they fail to show how they compare to wildtype mice of the same background, so this is a wrong statement. The body weight of the mice is not too hight indeed, but we need to have data of control mice next to them as a proof. Another negative aspect of the study is that these mice are not even littermates, one could have heterozygous breedings that result in all the desired genotypes, including the control wildtype mice, then the study would be complete. undesrstandably the authors cannot provide this but at least they should have control mice body weight on the same graph.
The insulin tolerance tests should also be provided as % of basal insulin, so they all start at 100% at the zero timepoint. Absolute glucose values sometimes can be misleading when it comes to insulin sensitivity.
Author Response
Comment: What is missing from the paper is any wildtype control mice as a reference. Especially when the authors claim that ldlr-/- and apoe-/- mice do not develop overweight when fed a western diet, yet they fail to show how they compare to wildtype mice of the same background, so this is a wrong statement. The body weight of the mice is not too hight indeed, but we need to have data of control mice next to them as a proof. Another negative aspect of the study is that these mice are not even littermates, one could have heterozygous breedings that result in all the desired genotypes, including the control wildtype mice, then the study would be complete. undesrstandably the authors cannot provide this but at least they should have control mice body weight on the same graph.
Response: We understand the reviewer’ comments. The hypothesis being tested is whether LDLR-/- mice develop hyperglycemia when fed a Western diet like Apoe-/- mice. Wild-type mice were not included as they only show modest rise in blood glucose levels and do not develop significant hyperglycemia (Scientific Reports 2019; 9: 19556. J Diabetes 2015;7:74. Parks et al., 2015, Cell Metabolism 21, 334). Our limited study of 4 female C3H/HeJ mice shows that these mice did not develop hyperplasia after 12 weeks on a Western diet (plasma levels of 148 ± 14 mg/dL) and nor overweight (21 ± 0.6 g), while female C3H-Apoe-/- mice had plasma glucose levels of 312 ± 31 mg/dL and body weight (21 ± 0.2 g) (Supplemental materials). We have addressed this concern in Discussion.
C3H-Apoe-/- and C3H-LDLR-/- mice are two different knockouts. The suggested littermates could be generated from the two knockouts within the time frame requested to submit this revision.
Comment: The insulin tolerance tests should also be provided as % of basal insulin, so they all start at 100% at the zero timepoint. Absolute glucose values sometimes can be misleading when it comes to insulin sensitivity.
Response: We have made the suggested amendment.
Reviewer 4 Report
The manuscript of “Ldlr-deficient mice develop severe hyperglycemia and type 2 diabetes on a Western-type diet” by W. Shi and co-authors aims to compare the effect of Western and chow diets on the development of type 2 diabetes in male and female C3H-Ldlr-/- and C3H-Apoe-/- mice. The research topic is interesting, but the manuscript contains several omissions and may be accepted for publication after major revision.
Comments:
1. The authors need to check the manuscript for compliance with the requirements of the Journal.
2. The Materials and Methods section: the chow diet should be described in more detail. The catalog number of ELISA kits used should be provided. It should be added to the section the manufacturer and the catalog number of insulin. It is necessary to clarify how the normality of the distribution was checked, and which tests were applied after ANOVA.
3. Fig. 1: There are no captions A and B. It is necessary to provide data on one axis (or to regroup the data as it was done in Fig. 3) and show statistical significance between different groups. According to the legend, there are 4 to 29 mice per group. Please explain such a large variation in the number of animals.
4. It is necessary to correct some conclusions if no statistically significant difference was found (For example, “Female C3H-Ldlr-/- mice also had a lower total cholesterol level than C3H-Apoe-/-, although the difference was not statistically significant”, etc.)
5. Fig. 2: Figures A and C should be combined. The same should be done for Figure B and D. This is due to the fact that the Results section discusses the differences between the groups that are presented in different figures.
6. Fig. 4.: It is necessary to calculate the area under the curve.
7. Some sentences need to be rewritten to make it sound more scientific (For example, “…Western diet consumption and ensuing hyperlipidemia lead to development of type 2 diabetes in hyperlipidemic mice, irrespective of its underlying causes”. It is necessary to add the Сonclusions section.
8. The Data availability section: There is no supplementary material file, as described.

Author Response
Comment 1. The authors need to check the manuscript for compliance with the requirements of the Journal.
Response: We have followed the instructions of the journal.
Comment 2. The Materials and Methods section: the chow diet should be described in more detail. The catalog number of ELISA kits used should be provided. It should be added to the section the manufacturer and the catalog number of insulin. It is necessary to clarify how the normality of the distribution was checked, and which tests were applied after ANOVA.
Response: Amended.
Comment 3. Fig. 1: There are no captions A and B. It is necessary to provide data on one axis (or to regroup the data as it was done in Fig. 3) and show statistical significance between different groups. According to the legend, there are 4 to 29 mice per group. Please explain such a large variation in the number of animals.
Response: The differences in plasma cholesterol levels between chow and Western diets are dramatic for both knockouts. The modest difference between two knockouts on the chow diet would be overwhelmed if plotted as Fig. 3). Due to small volumes of plasma achieved, some assays could be done for all mice.
Comment 4. It is necessary to correct some conclusions if no statistically significant difference was found (For example, “Female C3H-Ldlr-/- mice also had a lower total cholesterol level than C3H-Apoe-/-, although the difference was not statistically significant”, etc.)
Response: Revised.
Comment 5. Fig. 2: Figures A and C should be combined. The same should be done for Figure B and D. This is due to the fact that the Results section discusses the differences between the groups that are presented in different figures.
Response: The difference in cholesterol levels on the two diets is so dramatic that modest differences between the two knockouts in cholesterol levels on the chow diet would be covered if plotted in the same figure. We prefer the way as was.
Comment 6: Fig. 4.: It is necessary to calculate the area under the curve.
Response: Amended.
Comment 7. Some sentences need to be rewritten to make it sound more scientific (For example, “…Western diet consumption and ensuing hyperlipidemia lead to development of type 2 diabetes in hyperlipidemic mice, irrespective of its underlying causes”. It is necessary to add the Сonclusions section.
Response: Amended.
Comment 8. The Data availability section: There is no supplementary material file, as described.
Response: We have also made data accessible via Digital Object Identifier: 10.6084/m9.figshare.20032229
Round 2
Reviewer 2 Report
The authors have improved the article by including different data. To better understand and interpret the results they should indicate the number of samples included in each study and in each group (4 to 29 is a big difference and the study in only a few samples may limit the significance of the data shown).
The authors should include in the paper the definition of diabetes used.
Author Response
Comment: The authors have improved the article by including different data. To better understand and interpret the results they should indicate the number of samples included in each study and in each group (4 to 29 is a big difference and the study in only a few samples may limit the significance of the data shown).
Response: Amended. We repeated assays for groups with small animal number to ensure the reproducibility of the results.
Comment: The authors should include in the paper the definition of diabetes used.
Response: Amended
Reviewer 4 Report
The manuscript has been significantly improved; it can be accepted for publication in its current form.
Author Response
Comment: The manuscript has been significantly improved; it can be accepted for publication in its current form.
Response: We thank the reviewer for the comment.